# Investigation and Improvement of Test Methods for Capacitance and DCESR of EDLC Cells

**DOI:** 10.3390/s23104717

**Published:** 2023-05-12

**Authors:** Xiong Xie, Xu Li, Junqiang Xu, Lichun Dong

**Affiliations:** 1School of Chemistry and Chemical Engineering, Chongqing University, Chongqing 400044, China; xiongxie@zkcrtech.com; 2Chongqing CAS Supercap Technology Co., Ltd., Chongqing 401329, China; lxcqu@outlook.com; 3School of Chemistry and Chemical Engineering, Chongqing University of Technology, Chongqing 400054, China; xujunqiang@cqut.edu.cn; 4Chongqing Institute of Green and Intelligent Technology, Chinese Academy of Sciences, Chongqing 400714, China

**Keywords:** standard, test method, capacitance, DCESR, EDLC cells

## Abstract

The quick and accurate characterization of commercial electrochemical double-layer capacitor (EDLC) cells, especially their capacitance and direct-current equivalent series internal resistance (DCESR), is of great significance for the design, maintenance, and monitoring of EDLCs used in areas of energy, sensors, electric power, construction machinery, rail transit, automobile transportation, and military. In this study, the capacitance and DCESR of three commercial EDLC cells with similar performance were determined and compared by following the three commonly-used standards of IEC 62391, Maxwell, and QC/T741-2014, which are significantly different in test procedures and calculation methods. The analysis of the test procedures and results demonstrated that the IEC 62391 standard has the disadvantages of a large testing current, long testing time, and a complex and inaccurate DCESR calculation, whereas the Maxwell standard has the disadvantages of a large testing current, a small capacitance, and large DCESR testing results, and furthermore the QC/T 741 standard has the disadvantages of a high resolution requirement for the equipment and small DCESR results. Therefore, an improved method was proposed to determine the capacitance and DCESR of EDLC cells by short-time constant voltage charging and discharging interruption methods, respectively, with the advantages of high accuracy, low equipment requirements, short testing time, and the easy calculation of DCESR over the original three standards.

## 1. Introduction

Electrochemical double-layer capacitor (EDLC) cells are a type of emerging electrochemical energy storage device with a high power density of up to 15 kW/kg [1,2], and are widely used in the areas of energy, IoT sensors, wearable strain sensors, electric power, construction machinery, rail transit, automobile transportation, and military [2,3,4,5,6,7,8,9,10,11,12]. The accurate performance characterization of EDLC cells is of great engineering significance to ensure their safety and reliability, especially under the two critical parameters: the capacitance reflecting the capacity to store electric energy and the direct-current equivalent series internal resistance (DCESR) affecting their instantaneous discharge ability. In the EDLC packs used in electric power, rail transit, energy, or sensors, the capacitance and DCESR incompatibility between the composed cells would lead to the voltage imbalance during the charging and discharging process, resulting in the over-charging and over-discharging of one or multiple EDLC cells, which will ultimately affect the reliability and even the service life of the sensors [13]. Even for the sensors stored in the static state, the voltage difference between the composed EDLC cells could also cause the loss of stored energy, thereby reducing the usage time of sensors composed of EDLC packs. Therefore, accurately determining capacitance and DCESR is very helpful for taking suitable measures to ensure the safe operation of EDLC packs and even the sensors in their serving life.

In the literature, many studies have been carried out to accurately determine the capacitance and DCESR of EDLC cells, including that on test principles, models, and standards [14,15,16,17,18,19,20,21,22,23,24,25,26,27,28]. The typical test standards for commercial large-capacity EDLC cells are the IEC 62391-1 (International electrotechnical commission: Fixed electric double-layer capacitors for use in electronic equipment—Part 1: Generic specification) [29], the Maxwell standard 1007239 (Application note 1007239-Test Procedures for Capacitance, ESR, Leakage Current and Self-Discharge Characterizations of Ultracapacitors) [30], and the QC/T 741-2014 (the Chinese automotive industry standard: Ultra-capacitor for electric vehicles) [31], respectively, which are briefly referred to as the IEC 62391 standard, Maxwell standard, and the QC/T 741 standard in the following sections. On one hand, in the three standards, the determining procedures of capacitance and DCESR are all based on the principle of the classical constant current discharging method, in which, the test current of large-capacity EDLC cells is up to hundreds of amperes, putting forward high requirements for the test equipment for accurately controlling the charging and discharging current and the test voltage. Conversely, the test procedures and calculation methods for determining the capacitance and DCESR significantly differ in the three standards. Compared with that of the Maxwell standard and IEC 62391 standard, the test procedure of QC/T 741 standard is relatively easier and the corresponding testing current is relatively smaller; however, the testing time is much longer due to the small charging and discharging currents and a large number of cycles. Moreover, when using the QC/T 741 standard to determine the DCESR, the test equipment is required to have high-resolution accuracy for recording the voltage 30 ms after the start of the discharging process. Conversely, the Maxwell standard has the advantages of a simple program setting, short test time, and a low requirement for equipment resolution. However, the testing current is large (for 3000F EDLC cells, the test current is as high as 300 A). The test procedure of the IEC 62391 standard is obviously different, in which, (a). the EDLC cells are charged for 30 min with the constant rated voltage UR, greatly increasing the test time; (b). the testing current is also large (for 3000F EDLC cells, the test current is as high as 360 A). Consequently, there is no generally-recognized standard at home and abroad for determining the capacitance and DCESR of EDLC cells, which seriously affects the accurate evaluation of EDLC cells, and brings potential safety hazards to the use of EDLC cells in their later stage.

This study aims to develop an improved test method for quickly and accurately determining the capacitance and DCESR of EDLC cells by comparing and analyzing the previous test methods, including the procedures, calculation methods, and testing mechanisms in the three standards. Variation of the capacitance and DCESR values under different discharging currents and constant potential times were analyzed, and the reasons for different testing results were discussed. Subsequently, an improved method with a convenient test program setting, with low requirements for equipment accuracy, but that can accurately determine the capacitance and DCESR of large-capacity EDLC cells was proposed.

The remainder of the paper was organized as follows: In Section 2, the test equipment and procedure for determining the capacitance and DCESR in the three standards were compared, and their advantages and disadvantages were described. The capacitance and DCESR values for three commercial EDLC cells under the three standards and their evolution law under different and discharging currents and different constant voltage times were compared in Section 3, and the improved test method was proposed. Finally, the study was concluded in Section 4.

## 2. Experimental

### 2.1. EDLC Cells 

Three commercial EDLC cells that have a similar rated capacitance and DCESR were selected in the experiments [32,33,34]. They are representative of the EDLC technology with similar characteristics: (1). all cells have a cylindrical shape (which is the most common layout for EDLC cells), with the same diameter (60 mm) and length (138 mm); (2). the rated voltage ***U_R_*** and rated capacitance ***U_R_*** are similar for all three EDLC cells. The main parameters of the cells published by the manufacturer are listed in Table 1.

### 2.2. Test Equipment and Procedure

#### 2.2.1. LBT 20144 Test Equipment

The test equipment used in this study is the LBT 21044 high-precision battery test system produced by Arbin Instruments (College Station, TX, USA) with the technical parameters in Table 2. The LBT 21044 high-precision battery test system consists of the control software, power module, control module, and charging and discharging management module. The control module receives control information sent by the control software and distributes and transmits instructions through its Microcontroller Unit (MCU) section, and then converts them into digital analog (DA) signals to control the charging and discharging management module, enabling it to charge and discharge the EDLC cells according to the process requirements. The entire process is a closed-loop control structure; during the charging process, the power module provides energy and the EDLC cells are charged through the control of the charge and discharge management module. During the discharging process, the energy of the EDLC cells are absorbed and released by the charging and discharging management module. The measurement of data is achieved by collecting voltage integration and shunt resistor, and then integrated by the MCU module in time to obtain the required measurement data. This test system can detect the voltage, current, state of charge (SOC), and other states of various modules of the EDLC cells in real-time.

#### 2.2.2. Test Procedure

All tests were carried out at room temperature. Before starting the test, all EDLC cells were discharged to a voltage of 0.1 V with the same current to ensure that their initial state is consistent. The test procedure and calculation methods by the three standards, QC/T 741, Maxwell, and IEC 62391, are as follows. The QC/T 741 standard: (1). The EDLC cells are charged to the rated voltage ***U_R_*** with a constant current ***I***; (2). The cells are discharged to the minimum working voltage ***U_min_*** with a constant current ***I***; (3). Repeating the steps of 1 and 2 for 5 times, and the discharging time from **0.8*U_R_*** to the minimum working voltage ***U_min_*** is recorded as ***t***. The capacitance and DCESR are, then, calculated according to Equations (1) and (2). The average value of the capacitance of the five-cycle tests and that of the DCESR of the first three-cycle tests are recorded as the test values.
(1)$$C=\frac{I*t}{0.8U_{R}-U_{\min}}$$
(2)$$R=\frac{U_{R}-U_{i}}{2I}$$

The charging and discharging current are required to be 5***I*_1_** for energy-type cells and 40 ***I*_1_** for power-type cells, respectively (***I*_1_**= ***C_R_U_R_/*3600**). For the EDLC cells selected in this study, the maximum testing current is 100 A, which can be met by the test equipment.

The Maxwell standard: (1). The EDLC cells are rested for 10 s before the charging process; (2). The EDLC cells are charged to the rated voltage ***U_R_*** with a constant current ***I***; (3). Resting 5 s; (4). Resting 10 s; (5). The EDLC cells are discharged to half the rated voltage 0.5***U_R_*** with a constant current ***I***, (6). The EDLC cells are rested for another 5 s, and steps 1–6 are repeated twice. The voltages of ***U*_1_**, ***U*_2_**, and ***U*_3_** and the time ***t*_1_** and ***t*_2_** at the end of steps 4–6 are recorded, respectively. Finally, the EDLC cells are discharged to a terminal voltage of less than 0.1 V with a constant current ***I*** to ensure their safety [30]. The capacitance and DCESR can be calculated by Equations (3) and (4), respectively.
(3)$$C=\frac{I*\left(t_{2}-t_{1}\right)}{U_{1}-U_{2}}$$
(4)$$R=\frac{U_{3}-U_{2}}{I}$$

The charging and discharging current for the Maxwell standard are as high as 100 mA/F, for 3.0 V 3000 F EDLC cells, the corresponding test current is 300 A. In the low voltage range (0–5 V), few equipment can reach such a large testing current.

The IEC 62391 standard: (1). The EDLC cells are charged to the rated voltage ***U_R_*** with a constant current ***I_c_*** (***I_c_*** = **4*C_R_U_R_***) (2). The EDLC cells are charged for 30 min with the constant rated voltage ***U_R_***; (3). The EDLC cells are discharged to a terminal voltage of less than 0.1 V with a constant current ***I***. Then, the capacitance can be calculated by Equation (3). 

Where ***U*_1_** and ***U*_2_** were **0.8*U_R_*** and **0.4*U_R_***, respectively. The ***t*_1_** and ***t*_2_** are the times when the EDLC cells are discharged from the rated voltage ***U_R_*** to ***U*_1_** and ***U*_2_**, respectively. The discharging current varies with the type of EDLC cells. Taking EDLC cells of 3.0 V 3000F as examples, the discharging current for the memory backup type, energy type, power type, and instantaneous power type cells are 3 A, 3.6 A, 36 A, and 360 A, respectively, and the equipment cannot meet the requirements for the instantaneous power cells. The linear fitting of the discharging curve is employed to read the voltage drop **Δ*U*** between the fitting line and the discharging curve at the beginning of the discharging, and then, the DCESR value is calculated by Equation (5).
(5)$$R=\frac{△U}{I}$$

The maximum testing current required by the Maxwell standard and IEC 62391 standard for the large EDLC cells can reach 300 A and 360 A, respectively, which is challenging to be met by the LBT 20144 test equipment as the maximum current of the equipment is 100 A. Within the capability of the equipment, in order to explore the impact of the test current on the capacitance and DCESR for EDLC cells under three standards, the 50 A, 60 A, 70 A, 80 A, 90 A, and 100 A were selected as the testing current.

## 3. Results and Discussion

### 3.1. Comparative Analysis of the Three Test Standards

The detailed current and voltage waveforms during the test of the EDLC cells (3.0 V 3000F, CAS SCAP) according to the three standards were illustrated in Figure 1, showing that all the three standards adopt the procedure of constant current charging to the rated voltage ***U_R_***, followed by constant current discharging to half of the rated voltage 0.5***U_R_*** or a terminal voltage of 0.1 V based on test standard requirements. Compared with that of the Maxwell standard and IEC 62391 standard, the test procedure of the QC/T 741 standard is relatively easier and the corresponding testing current is relatively smaller; however, the testing time is much longer due to the small charging and discharging currents and a large number of cycles. Moreover, when using the QC/T 741 standard to determine the DCESR, recording the voltage 30 ms after the start of the discharging process required the test equipment to have a high-resolution accuracy. Conversely, the Maxwell standard had the advantages of a simple program setting, short test time, and low requirement for equipment resolution. However, the testing current is large (for 3000F EDLC cells, the test current was as high as 300 A), which is difficult for general commercial equipment to meet. The test procedures of the IEC 62391 standard are obviously different, in which, (a). the EDLC cells are charged for 30 min with the constant rated voltage ***U_R_***, greatly increasing the test time; (b). the testing current is also large (for 3000F EDLC cells, the test current is as high as 360 A).

Table 3 showed the as-obtained capacitance for the SCD3R0308L29CAZ EDLC cell according to the three standards in different voltage ranges, showing that the values calculated using different voltage ranges for the same test are significantly different. For the same EDLC cell under the same voltage ranges, the capacitance value determined according to the IEC standard was the largest, and that according to the QC/T 741 standard was the smallest. The maximum deviation is around 1.22%. Moreover, it could be seen from the results according to the IEC 62391 standard that the larger the voltage range or the farther it deviates from the initial discharging voltage, the smaller the calculated capacitance value is, which was consistent with the study of Guo et al. [35]. Therefore, when determining the capacitance, the calculated value should be taken as close to the line segment at the beginning of the discharging as possible.

Figure 2 showed the diagram of the voltage drop **∆*U*** on the discharging curve under the three standards. At the beginning and end of the constant current charging and discharging process, voltage drop **∆*U*** originating from DCESR will be generated in the discharging curve. Therefore, accurate measurement of the voltage drop **∆*U*** is the key to accurate calculation of DCESR. It can be seen from Figure 2a that, according to the QC/T 741 standard, the discharging curve did not reach the linear region after starting the discharge for 30 ms, and therefore the voltage drop **∆*U*** caused by DCESR has not been fully reflected through the discharging curve within 30 ms. Therefore, the DCESR value obtained according to the discharging curve at this stage cannot reflect the true value of the EDLC cells and cause the determined DCESR value to be small. The discharging curve of the Maxwell standard (Figure 2b) cannot reflect the voltage drop in detail due to a large recording time interval, which can only be calculated according to the voltage at 5 s after the discharging is stopped. The Maxwell standard discharging curve (Figure 2b) has a large measured value of DCESR due to the large recording time interval and the excessive count of the voltage drop **∆*U***. For the IEC 62391 standard, the calculation of voltage drop **∆*U*** is significantly dependent on the data segments selected for fitting, which inevitably leads to subjective errors. Therefore, it is error-prone to measure the DCESR values according to the IEC 62391 standard, therefore the calculation and determination of DCESR by using the IEC 62391 standard is not discussed in this study.

### 3.2. Comparison of Test Results 

The failure curve of the EDLC follows the ‘bathtub curve’ [17]. That is, during the initial cycle stage, due to the instability of materials (the rapid decay of pseudocapacitance caused by oxygen containing functional groups, impurities, etc.), the capacitance decreases rapidly. After several cycles, the capacitance attenuation slows down and the DCESR basically remains unchanged from the beginning of the cycle. Because supercapacitors are physical devices that do not undergo chemical reactions during energy storage and transfer, their performance remains basically unchanged for a short time/a few cycles. Figure 3 verified that the capacitance of three commercial EDLC cells decrease gradually with an increase in cycles, while the DCESR values basically remain constant in the initial cycle. More importantly, the capacitance and DCESR values obtained according to the Maxwell standard are much larger than those obtained according to the QC/T 741 standard. Take data of the second cycle of SCD3R0R308L29CAZ as an example, the difference of the capacitance and DCESR values can reach as large as 75.4 F and 0.075 mΩ, respectively.

Figure 4 demonstrated that, under the IEC 62391 standard, the capacitance of the EDLC cells firstly increased gradually with an increase in the constant voltage charging time, reaching a basically unchanging value after the constant voltage charging time exceeded 1 min. Moreover, for all three EDLC cells, the capacitance and DCESR according to different standards were significantly different. In terms of capacitance, the values obtained according to the IEC 62391 standard were the largest, while those obtained according to the QC/T 741 standard were the smallest, as seen in Figure 3 and Figure 4.

The capacitance of the EDLC cells mainly depends on the adsorption and desorption of ions in their electrodes [36]. A charging procedure at a small current is helpful to the transfer of ions between the electrodes and promoted the EDLC cells to quickly reach the fully charged state. Figure 4 showed that the charging current of ***I*** = **4*C_R_U_R_*** and the constant voltage charging time of 1 min have met the time required for the complete adsorption of ions on the electrode surface. So, the capacitance of the EDLC cells kept a basically unchanging value after the constant voltage charging time exceeded 1 min. In addition, according to the energy storage mechanism of the double layer, the ions will continue to move towards the electrode and be adsorbed under constant voltage and resting state after EDLC cells have been charged to the rated voltage, and more ions can be adsorbed under constant voltage than under resting in EDLC cells. Therefore, the capacitance value obtained according to the IEC 62391 standard was larger than that according to Maxwell standards and both were greater than those obtained according to QC/T 741 standard. Moreover, according to the statistical results in Table 3, the calculated capacitance values for the voltage range from ***U_R_*** to **0.5*U_R_*** were greater than those calculated from **0.8*U_R_*** to the minimum discharging voltage ***U_min_***. According to the IEC 62391 and Maxwell standards, the selected voltage range is from ***U_R_*** to **0.5*U_R_***, while the voltage range of the QC/T 741 standard is from **0.8*U_R_*** to ***U_min_***. Consequently, the capacitance values according to the QC/T 741 standard were lower than those obtained under the IEC 62391 and Maxwell standards. However, in the Maxwell standard, EDLC cells underwent a 15 s resting process before discharging process. Due to the influence of internal resistance and leakage current, the voltage would decrease within 15 s, and the discharging starting voltage was slightly lower than the rated voltage. Therefore, the capacitance values according to the Maxwell standard were lower than those obtained under the IEC 62391 standard.

Figure 5 showed the capacitance and DCESR of the three commercial EDLC cells tested and calculated at different charging-discharging currents according to different standards. The capacitance value obtained according to the IEC 62391 standard was slightly larger than that according to Maxwell standards and larger than that according to QC/T 741 standards at all testing currents, which were consistent with the results in Figure 3 and Figure 4. Moreover, capacitance varied with the testing current under all three standards, and performance varied under different standards. The capacitance values obtained according to the IEC 62391 standard remained unchanged with an increase in test current, whereas those obtained according to the Maxwell standard deceased significantly with an increase in testing current, and furthermore the capacitance values obtained according to the QC/T 741 standard increased with an increase in testing current. As for the DCESR, the values obtained under the Maxwell standard were much larger than those according to the QC/T 741 standard; moreover, the DCESR values under the Maxwell standard decreased with an increase in the testing current, while those under the QC/T 741 standard remained basically invariant at different currents. 

With an increased discharging current, the capacitance values obtained under the IEC 62391 standard remain unchanged. The reason is that the anions and cations adsorbed at the positive and negative electrodes reach a saturation state under the constant voltage state; consequently, the capacitance would not vary with the test current since the ions released from the electrodes do not change with the test current. While during the resting state, after the EDLC cells are fully charged in the Maxwell standard, some anions and cations near the electrodes can continue to move to the positive and negative electrodes. Moreover, a charging procedure at a small current is helpful to the transfer of ions between the electrodes, promoting the EDLC cells to quickly reach the fully charged state; consequently, more ions can move to the electrodes at a smaller charging current, and more ions can be released form the electrodes at a smaller discharging current, resulting in smaller capacitance values at larger charging and discharging currents. On the other side, under the QC/T 741 standard, the EDLC cells are discharged directly after they are charged without the constant voltage or resting process, and there is no intermittent time for ions to continue moving toward the electrodes after the charging process. Therefore, more ions can be adsorbed and released from the electrodes in the charging and discharging processes, respectively, resulting in larger capacitance values at larger charging and discharging currents under the QC/T 741 standard. 

The internal resistance of EDLC cells, i.e., DCESR in the classical equivalent model, results in the migration of ions in the electrolyte and membrane, and the adsorption and desorption of ions on or from the electrode [16]. According to the double layer theory, the ions in EDLC cells become solvated ions by absorbing solvents via electrostatic effects, which migrate in electrolytes and membranes and are adsorbed or released in the macropores of activated carbon of electrodes. However, in the small pores on the surface of electrodes, the adsorption and desorption of solvated ions are relatively difficult, often accompanied by de-solvation. However, the de-solvation of ions consumes energy, which is also manifested as an increase in DCESR of EDLC cells [16]. According to the QC/T 741 standard, the DCESR values are determined by using the voltage drop **∆*U*** from the beginning of the discharging process to the time of 30 ms, leading to relatively smaller DCESR values due to the incomplete ion migration, desorption, and de-solvation. Moreover, in such a short time, the DCESR mainly results from ion de-solvation and desorption, which is basically not affected by the discharging current. On the contrary, in the Maxwell standard, the DCESR values are calculated according to the voltage difference between the voltage at 5 s after the discharge was interrupted and the half-rated voltage. In such a long time, the migration, desorption, and de-solvation of ions can be completely fulfilled, leading to relatively larger DCESR values since the resistance of ions in the diffusion layer migrating in the membrane and electrolyte is also considered. Moreover, the resistance of ion migration decreases with an increase in test current, therefore, the DCESR values obtained according to the Maxwell standard decrease with an increase in testing current.

### 3.3. Improvement of the Test Method

The above analysis demonstrated that all three standards have shortcomings, i.e., the IEC 62391 standard has the disadvantages of a large testing current, long testing time, and complex and inaccurate DCESR calculation, whereas the Maxwell standard has the disadvantages of a large testing current, small capacitance, and large DCESR results, and furthermore the QC/T 741 standard has the disadvantages of a high equipment resolution requirement and small DCESR results. Accordingly, an improved method is proposed in this study with the following considerations: (1). EDLC cells should be charged at a constant voltage for a period of time before being discharged to achieve a fully charged state before discharging, thereby obtaining more accurate capacitance results; (2). The discharging curve between rated voltage ***U_R_*** and **0.5*U_R_*** should be used to calculate capacitance to eliminate the impact of the internal resistance and leakage current, leading to more accurate capacitance results; (3). the method of calculating the DCESR value by the voltage difference between the recovery voltage and the **0.5*U_R_*** use in the Maxwell standard can achieve the observation of the change rule of the recovery voltage. This method can clarify which difference between the voltage recovery at some point and the **0.5*U_R_*** represents the DCESR value of the EDLC cells. Consequently, the procedure for measuring the capacitance and DCESR of EDLC cells is as follows: (1). The EDLC cells are rested for 10 s; (2). The EDLC cells are discharged to a voltage of less than 0.1 V; (3). The EDLC cells are charged to the rated voltage ***U_R_*** with a constant current ***I*** (***I*** = **4*C_R_U_R_***); (4). The EDLC cells are charged for 1 min with the constant rated voltage ***U_R_***; (5). The EDLC cells are discharged to half of the rated voltage **0.5*U_R_*** with a constant current ***I*** (***I*** = 50 A, 60 A, 70 A, 80 A, 90 A, 100 A, respectively); (6). The EDLC cells are rested for 5 s; (7). The EDLC cells are discharged to a terminal voltage of less than 0.1 V with a constant current ***I*** (***I*** = 50 A) to ensure safety [30]. In the process, the voltage ***U*_1_**, ***U*_2_**, ***U*_3_** at the end of steps 4–6 and the time ***t*_1_**, ***t*_2_** at the end of steps 5–6 was recorded. The capacitance and DCESR of the EDLC cells can then be calculated according to Equations (3) and (4), and the variation of the voltage and current during the characterization of EDLC cells according to the improved test method was illustrated in Figure 6a.

By using the improved test method, the capacitance and DCESR of the three commercial EDLC cells were tested under different discharging currents. The results in Figure 6b demonstrate that the capacitance of the EDLC cells remains almost invariant with different discharging currents. According to the charge storage mechanism, a charging procedure at a small current and then a constant voltage is helpful to the transfer of ions between the electrodes, promoting the EDLC cells to quickly reach the fully charged state. When the EDLC cells continue to be charged at the constant voltage after they were fully charged, the anions and cations continue to move towards and be adsorbed on the electrodes to reach the saturation state. While in the discharging state, the ions adsorbed on the electrodes can be completely released, which was independent of the discharging current, resulting in a higher capacitance value.

In the improved method, the DCESR of EDLC cells was tested and calculated via a discharging interruption approach. If a discharging process is suddenly interrupted, the current quickly drops to zero (***I*** = 0), while the voltage recovers immediately, the DCESR of the tested EDLC cells can then be calculated by dividing the discharging current by the voltage difference between the recovered voltage and the voltage at the time of the discharging interruption. The discharging interruption approach for determining the DCESR values of EDLC cells has the advantages of simple procedures and low equipment resolution requirements. Therefore, in most cases, the discharging interruption approach is much more convenient than the initial current method [36]. In the above standards, the Maxwell standard uses the discharging interruption approach, while the IEC 62391 standard and the QC/T 741 standard use the initial current method.

The voltage-recovery curves of the three EDLC cells under different discharging currents were shown in Figure 7a–c, demonstrating the process of the EDLC cell’s voltage recovering from half of the rated voltage with the resting time after the discharging is suddenly interrupted. The calculated values, ***R*** = 1000**Δ*U***/***I***, of the three EDLC cells versus the resting time within 0.5 s after the discharging interruption were shown in Figure 7d,e. It can be seen that after the sudden interruption, the ***R*** values of the tested EDLC cells first increase rapidly with the rest time, while after about 200 ms, the increasing rate slow down and the ***R*** values gradually reach a steady state.

After the discharging interruption, the recovery of EDLC voltage comes from the migration of solvated ions in the electrolyte and membrane and their adsorption on the electrode surface. When the discharging process is suddenly interrupted, the solvated ions that were released during the discharging process could be moved back from the electrolyte to the electrodes due to the electrostatic interaction, causing a sudden increase of the EDLC’s voltage. As the time went on, the electrostatic adsorption of the ions in the electrolyte on the electrode gradually decreases and the growth of the EDLC’s voltage slows down. Moreover, the migration of solvated ions adsorbed on the electrode surface to the interior of the electrode is also accompanied by a de-solvation process, which also takes time. It is generally considered that the ion readsorption process can be completed in 200 ms after the discharging interruption and the calculated DCESR value at this time is more accurate. This observation is verified by the results in Figure 7d,e, showing that the calculated DCESR values gradually flatten out after about 200 ms of discharging interruption. Different from the Maxwell standard, the DCESR value determined according to the improved method eliminates the resistance of ions near the membrane to move between the electrolyte and the membrane in the diffusion layer after the end of the discharge. Therefore, they are smaller but more reasonable than the values obtained through the Maxwell standard.

Figure 8 compared the capacitance and DCESR values of the three EDLCs that are obtained according to the improved test method with those obtained according to the three standards. The capacitance values determined by the improved method were close to those by the IEC 62391 standard, larger than those obtained by the Maxwell standard and QC/T 741 standard. The DCESR values measured by the improved method are between those obtained by the Maxwell standard and the QC/T 741 standard. Moreover, the values of capacitance and DCESR obtained by the improved method meet the indexes in the specification and keep stable with the change of the test current, as seen in Figure 6b and Figure 7d,e.

## 4. Conclusions

In this study, the capacitance and DCESR of three commercial EDLC cells with similar performances were determined and compared by following the three commonly-used standards of the IEC 62391, Maxwell, and QC/T741-2014, which are significantly different in test procedures and calculation methods. The analysis of the test procedures and results demonstrated that the IEC 62391 standard has the disadvantages of a large testing current, long testing time, and complex and unclear DCESR calculation methods; the Maxwell standard has shortcomings such as a high testing current, low capacitance testing results, and high DCESR testing results; the QC/T 741 standard has the disadvantages of a high requirement for equipment resolution and small DCESR testing results. 

On the other side, it was also found that the low current and short time constant voltage charging process in the IEC 62391 standard is helpful to the complete adsorption of ions on the surface of EDLC’s electrode during the testing process, making the EDLC stably reach the fully charged state. Therefore, the measured capacitance values according to the IEC 62391 standard are relatively large and do not change with the discharging current. Furthermore, the discharging interruption method used in the Maxwell standard is easy for recording the recovered voltage, which is conducive to the clarity of the test program and calculation results of EDLC’s DCESR values.

According to the above analysis, an improved method for measuring the capacitance and DCESR of EDLCs is proposed, in which, a small current constant voltage charging approach is used to test the capacitance, and a discharging interruption method is used to determine the DCESR values. The test results demonstrated that the capacitance and DCESR values determined by the improved test method do not change with the change of discharging currents. The capacitance values determined by the improved method are consistent with those measured by the IEC 62391 standard, larger than those by the Maxwell standard and the QC/T 741 standard. The DCESRs determined by the improved method are between the values obtained by the Maxwell standard and the QC/T 741 standard. Moreover, the test procedure of the improved method is simple, the requirements for recording data points are clear, and the test results are not affected by the discharging currents; therefore, the results measured by the improved method are more accurate than those determined by the existing standards.

## Figures and Tables

**Figure 1 sensors-23-04717-f001:**
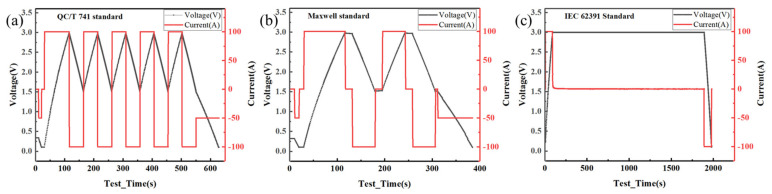
Current and voltage waveforms during the test of EDLC cells according to the (**a**) QC/T 741 standard, (**b**) Maxwell standard, and (**c**) IEC 62391 standard.

**Figure 2 sensors-23-04717-f002:**
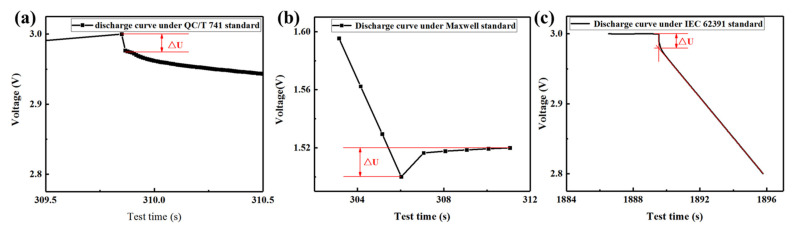
Voltage drop **Δ*U*** of the discharging curve according to the three standards. (**a**) QC/T 741 standard, (**b**) Maxwell standard, and (**c**) IEC 62391 standard.

**Figure 3 sensors-23-04717-f003:**
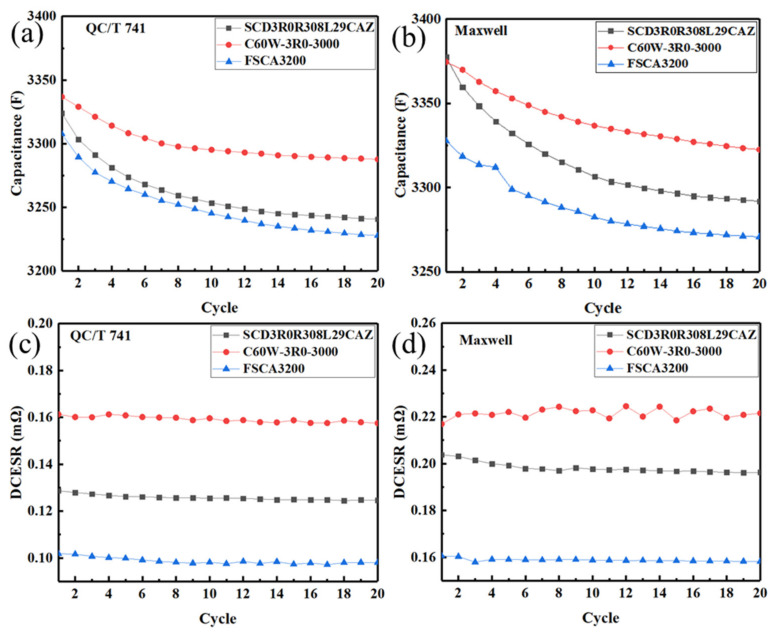
Variation of capacitance and DCESR of three EDLC cells with cycles under the QC/T 741 and Maxwell standards. The variation of capacitance under the QC/T 741 standards (**a**) and Maxwell standards (**b**), The variation of DCESR under the QC/T 741 standards (**c**) and Maxwell standards (**d**).

**Figure 4 sensors-23-04717-f004:**
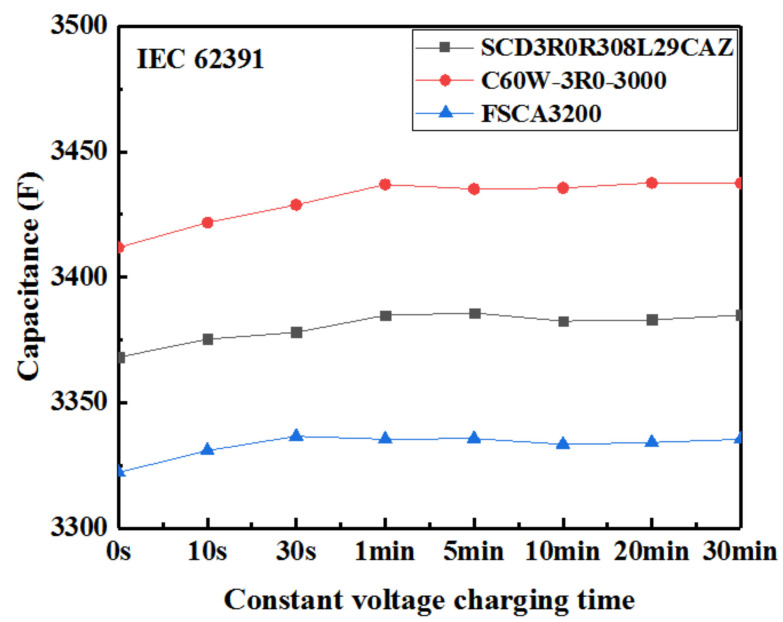
Variation of capacitance of three EDLC cells with the constant voltage charging time under IEC 62391 standard.

**Figure 5 sensors-23-04717-f005:**
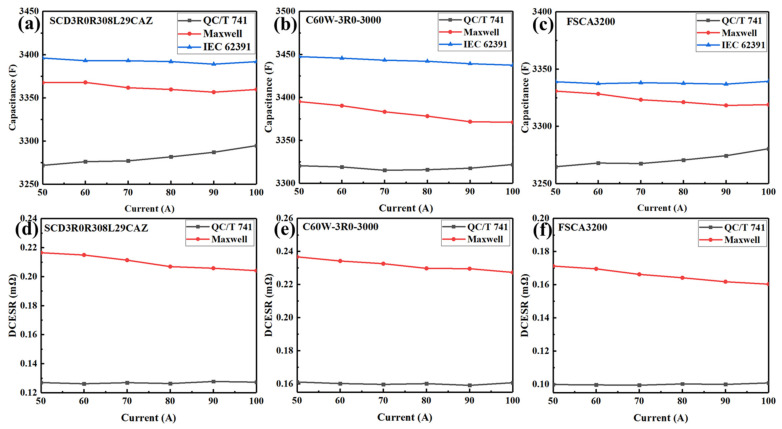
Capacitance and DCESR of the three large-capacity EDLC cells of 3000F at different charging-discharging currents. (**a**–**c**): Capacitance obtained according to the three standards. (**d**–**f**): DCESR obtained according to the QC/T 741 and maxwell standards.

**Figure 6 sensors-23-04717-f006:**
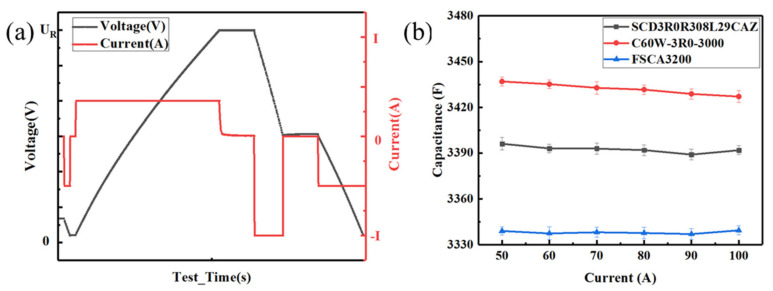
(**a**) Variation of the voltage and current during the characterization of EDLC cells according to the improved test method. (**b**) Variation of the capacitance with discharging current under the improved test method.

**Figure 7 sensors-23-04717-f007:**
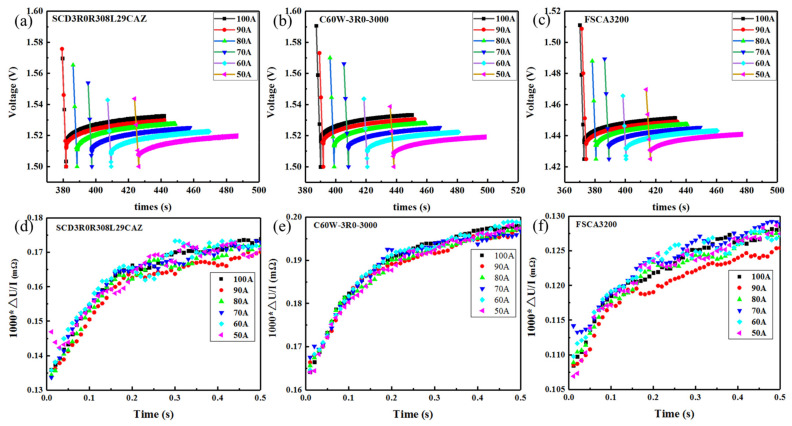
The voltage-recovery curves of the three EDLC cells under different discharging currents via the discharging interruption approach. (**a**) SCD3R0R308L29CAZ 3.0 V 3000F cell, (**b**) C60W-3R0-3000 3.0 V 3000F cell, (**c**) FSCA3200 2.85 V 3200F cell; The calculated values ***R*** = 1000**Δ*U***/***I*** of the three EDLC cells versus resting time within 0.5 s after discharging interruption (**d**) SCD3R0R308L29CAZ 3.0 V 3000F cell, (**e**) C60W-3R0-3000 3.0 V 3000F cell, (**f**) FSCA3200 2.85 V 3200F cell.

**Figure 8 sensors-23-04717-f008:**
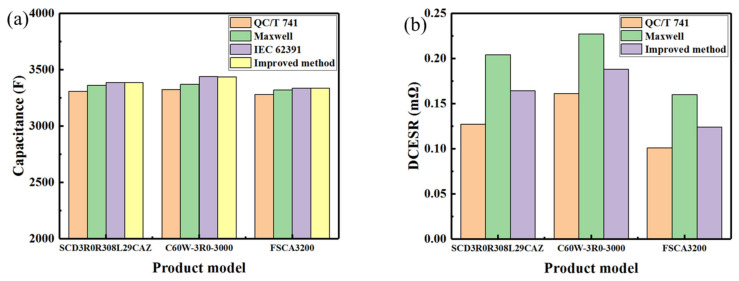
The value of capacitance and DCESR for three commercial EDLC cells according to the three standards and improved method. (**a**) Value of capacitance, (**b**) Value of DCESR.

**Table 1 sensors-23-04717-t001:** Technical parameters of the three commercial EDLC cells.

Manufacturer	CAS SCAP [32]	GMCC [33]	SKELETON [34]
Product model	SCD3R0308L29CAZ	C60W-3R0-3000	FSCA3200
Rated Voltage (***U_R_***)	3.0 V	3.0 V	2.85 V
Rated capacitance (***C_R_***)	3000 F	3000 F	3200 F
Rated DCESR	0.23 mΩ	0.18 mΩ	0.18 mΩ
Rated continuous current (I_r_ at ∆T = 15 °C)	144 A	164 A	190 A
Overall Diameter	60.5 mm	60 mm	60.2 mm
Overall Length	138 mm	138 mm	138 mm

**Table 2 sensors-23-04717-t002:** Technical parameters of LBT 20144 test equipment.

Model	LBT 21044
Voltage control range	0–5 V
Standard current range	100 A/10 A/1 A/0.1 A
Test accuracy	±0.02%
Resolution ratio	24 bit
Minimum voltage @ maximum current	0 V @ 100 A
Number of standard channels	8

**Table 3 sensors-23-04717-t003:** Capacitance for SCD3R0308L29CAZ EDLC cells according to the three standards in different voltage ranges.

Capacitance (F)	*U_R_*~0.5*U_R_*	0.8*U_R_*~0.5*U_R_*	0.8*U_R_*~0.4*U_R_*	*U_R_*~0.1*V*	0.8*U_R_*~0.1*V*
QC/T 741 standard	3356.1	3306.1	/	/	/
Maxwell standard	3359.6	3305.7	/	/	/
IEC 62391 standard	3384.8	3331.0	3265.2	3111.7	3016.5

## Data Availability

Not applicable.

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
