# Peer review of "Investigation and Improvement of Test Methods for Capacitance and DCESR of EDLC Cells"

_sensors, 2023, doi:10.3390/s23104717_

Round 1
Reviewer 1 Report
Reviewed article described EDLC cells and method of determination of capacitance and DCESR. Paper is well prepared and could be published after minor revisions, listed below:
1. Table 3. Provide unit for capacitance values.
2. Figure 7 d-f. provide units on y axis
3. In conlusions there are two various fonts used.
4. Section 2.2.2 Would you describe in details equipment, that you used for tests?
5. In introduction. Could you explain differences in standards, what you used for tests.
I have no comments
Author Response
Lichun Dong, Ph.D
School of Chemistry and Chemical Engineering, Chongqing University,
Chongqing, 400044, China
E-mail: lcdong72@cqu.edu.cn
Tel.: +86 13640514654
May 5, 2023
Dear Editor and Reviewers:
Thank you for your comments and suggestions concerning our manuscript entitled “Investigation and improvement of test methods for capacitance and DCESR of EDLC cells” (ID: sensors-2356522), which are very helpful for us to revise the manuscript, as well as our further research work. According to the comments, we carefully revised the manuscript (the revised contents are highlighted in red color in the manuscript), and point-to-point responses are as follows.
Sincerely,
Lichun Dong
Point-to-point responses to reviewers’ comments
Reviewer #1:
Reviewed article described EDLC cells and method of determination of capacitance and DCESR. Paper is well prepared and could be published after minor revisions, listed below:
- Table 3. Provide unit for capacitance values.
Response: Thanks for your careful examination, and we are very sorry for this mistake. The missing unit have been added in Table 3 and marked in yellow for easy viewing.
- Figure 7 d-f. provide units on y axis
Response: Thanks. The missing units have added in Figure 7 d-f.
- In conlusions there are two various fonts used.
Response: We appreciate this good comment. In the revised manuscript, we adjusted the font to be consistent in the conclusion section, and also checked the fonts of the entire manuscript.
Section 2.2.2 Would you describe in details equipment, that you used for tests?
Response: Thanks for this good suggestion. In the revised manuscript, we have added a detail description of the test equipment in Section 2.2.1.
- In introduction. Could you explain differences in standards, what you used for tests.
Response: According to the suggestion, we have added a description of the differences between the three standards in introduction, mainly as follows:
Compared with that of the Maxwell standard and IEC 62391 standard, the test procedure of QC/T 741 standard is relatively easier and the corresponding testing current is relatively smaller; however, the testing time is much longer due to the small charging and discharging currents and a large number of cycles. Moreover, when using the QC/T 741 standard to determine DCESR, recording the voltage 30 ms after the start of the discharging process required the test equipment to have high-resolution accuracy. Conversely, the Maxwell standard had the advantages of a simple program setting, short test time, and low requirement for equipment resolution. However, the testing current is large (for 3000F EDLC cells, the test current was as high as 300A). The test procedures of the IEC 62391 standard are obviously different, in which, a). the EDLC cells are charged for 30 min with the constant rated voltage UR, greatly increasing the test time; b). the testing current is also large (for 3000F EDLC cells, the test current is as high as 360A). In addition, by comparing and analyzing the test method, including the procedures, calculation methods, and testing mechanism in the three standards. We have analyzed variation of the capacitance and DCESR values under different dis-charging currents and constant potential times, and the reasons for different testing results were also discussed. Subsequently, an improved method to determine the capacitance and DCESR of large-capacity EDLC cells was proposed.
Reviewer 2 Report
Report on: “Investigation and improvement of test methods for capacitance and DCESR of EDLC cells”, X. Xie et al.
Ref. Sensors 2356522.
General considerations:
The manuscript describes a study on the performance of three commercial devices for the estimate of the direct-current equivalent series internal resistance (DCESR) of electrochemical double-layer capacitor (EDLC) cells, an essential parameter to evaluate their capabilities as energy storage devices. The manuscript presents an interesting study although accompanied by several weaknesses needing revision, as detailed below.
Remarks:
1) The authors use a commercial “high-precision battery test” (see Experimental section 2.2). It is unclear
II) The authors provide in Table 3 a series of capacitance values.
a) Units are absent. Are non-dimensional quantities tabulated here?
b) The number of significant figures seems unrealistic. If the test accuracy is of 0.02 % (Table 2), the standard deviation of the reported quantities should be at least of 0.7 units. Probably, this standard deviation will be larger if replicate tests are carried out. In short, the differences in the values of the quantities in the two first columns of Table 3 will be, probably, non significantly statistically different. The discussion in page 5 around data in Table 3 is probably irrelevant.
III) The same problem appears in the analysis of data in Figures 3-5. These apparently correspond to a unique test. Logically, replicate tests should be employed and in these graphs the data points should be replaced by error bars.
Author Response
Lichun Dong, Ph.D
School of Chemistry and Chemical Engineering, Chongqing University,
Chongqing, 400044, China
E-mail: lcdong72@cqu.edu.cn
Tel.: +86 13640514654
May 5, 2023
Dear Editor and Reviewers:
Thank you for your comments and suggestions concerning our manuscript entitled “Investigation and improvement of test methods for capacitance and DCESR of EDLC cells” (ID: sensors-2356522), which are very helpful for us to revise the manuscript, as well as our further research work. According to the comments, we carefully revised the manuscript (the revised contents are highlighted in red color in the manuscript), and point-to-point responses are as follows.
Sincerely,
Lichun Dong
Point-to-point responses to reviewers’ comments
Reviewer #2:
The manuscript describes a study on the performance of three commercial devices for the estimate of the direct-current equivalent series internal resistance (DCESR) of electrochemical double-layer capacitor (EDLC) cells, an essential parameter to evaluate their capabilities as energy storage devices. The manuscript presents an interesting study although accompanied by several weaknesses needing revision, as detailed below.
Remarks:
1) The authors use a commercial “high-precision battery test” (see Experimental section 2.2). It is unclear
Response: We appreciate this good comment. In the revised manuscript, a detail description of the test system for EDLC cells is added in Section 2.2.1 to clarify the “high-precision battery test”.
2) The authors provide in Table 3 a series of capacitance values.
- a) Units are absent. Are non-dimensional quantities tabulated here?
Response: Thanks for this careful examination. According to the comment, the missing unit have added in Table 3 and marked in yellow for easy viewing.
- b) The number of significant figures seems unrealistic. If the test accuracy is of 0.02 % (Table 2), the standard deviation of the reported quantities should be at least of 0.7 units.Probably, this standard deviation will be larger if replicate tests are carried out. In short, the differences in the values of the quantities in the two first columns of Table 3 will be, probably, non significantly statistically different. The discussion in page 5 around data in Table 3 is probably irrelevant.
Response: Thanks for this excellent comment. The difference in the capacitance values obtained under different standards and voltage ranges listed in Table 3 is mostly very significant, tens or even hundreds of Faradas, only two data stand out as 3306.1 F vs 3305.7 F (the third column). As the reviewer suggested, the standard deviation of the capacitance values is around 0.7 F. Therefore, most of the reported data are statistically significant. Moreover, the reported data are the average value of several replicate tests, therefore, the standard deviation of the replicate tests is already considered.
3) The same problem appears in the analysis of data in Figures 3-5. These apparently correspond to a unique test. Logically, replicate tests should be employed and in these graphs the data points should be replaced by error bars.
Response:
Figure 3 show the variation of capacitance and DCESR of three EDLC cells with cycles under the QC/T 741 and Maxwell standards, while Figure 4 show the variation of capacitance of three EDLC cells with the constant voltage charging time under IEC 62391 standard. The focus of the discussion is to compare and analyze the effects of cycles and constant voltage times on the test results in a single test under a same testing standard. Therefore, there is no need to consider the difference of the values of the replicate tests. It is similar for Figure 6a, which shows the variation of the voltage and current during the characterization of EDLC cells according to the improved test method.
Figure 5 shows capacitance and DCESR values of the three large-capacity EDLC cells at different charging-discharging currents under three standards. Figure 6b shows the variation of the capacitance with discharging current under the improved test method. According to the suggestion of the reviewer, for demonstrating the deviation of replicate tests, replicate tests were employed and in these two Figures, and the average values with error bars were reported.
Round 2
Reviewer 2 Report
The manuscript can be published in its current version.